# Automatic Hyoid Bone Tracking in Real-Time Ultrasound Swallowing Videos Using Deep Learning Based and Correlation Filter Based Trackers

**DOI:** 10.3390/s21113712

**Published:** 2021-05-26

**Authors:** Shurui Feng, Queenie-Tsung-Kwan Shea, Kwok-Yan Ng, Cheuk-Ning Tang, Elaine Kwong, Yongping Zheng

**Affiliations:** 1Department of Biomedical Engineering, The Hong Kong Polytechnic University, Hung Hom, Kowloon 999077, Hong Kong; suri.d.feng@connect.polyu.hk (S.F.); queenie.tk.shea@connect.polyu.hk (Q.-T.-K.S.); 2Department of Chinese and Bilingual Studies, The Hong Kong Polytechnic University, Hung Hom, Kowloon 999077, Hong Kong; kwok-yan.ng@connect.polyu.hk (K.-Y.N.); leah-cn.tang@connect.polyu.hk (C.-N.T.)

**Keywords:** tracking, deep learning, correlation filters, dysphagia, ultrasound videos, hyoid bone, swallowing, SiamFC, real-time

## Abstract

(1) Background: Ultrasound provides a radiation-free and portable method for assessing swallowing. Hyoid bone locations and displacements are often used as important indicators for the evaluation of swallowing disorders. However, this requires clinicians to spend a great deal of time reviewing the ultrasound images. (2) Methods: In this study, we applied tracking algorithms based on deep learning and correlation filters to detect hyoid locations in ultrasound videos collected during swallowing. Fifty videos were collected from 10 young, healthy subjects for training, evaluation, and testing of the trackers. (3) Results: The best performing deep learning algorithm, Fully-Convolutional Siamese Networks (SiamFC), proved to have reliable performance in getting accurate hyoid bone locations from each frame of the swallowing ultrasound videos. While having a real-time frame rate (175 fps) when running on an RTX 2060, SiamFC also achieved a precision of 98.9% at the threshold of 10 pixels (3.25 mm) and 80.5% at the threshold of 5 pixels (1.63 mm). The tracker’s root-mean-square error and average error were 3.9 pixels (1.27 mm) and 3.3 pixels (1.07 mm), respectively. (4) Conclusions: Our results pave the way for real-time automatic tracking of the hyoid bone in ultrasound videos for swallowing assessment.

## 1. Introduction

Swallowing problems, also called dysphagia, have a prevalence of 16–23% in the general population, reaching 27% in those over 76 years of age [1]. It influences 16% of 87 years or older group in the Netherlands [2], affecting up to 40% of people in permanent-care settings [3] and 50 to 75% of nursing home residents [4]. Martino et al. (2005) reported that up to 37–78% of stroke patients have dysphagia [5]. Sapir et al. (2008) demonstrated that 90% of Parkinson’s disease patients present with dysphagia [6].

The clinical gold standard for swallowing disorder assessments is the videofluoroscopy swallowing study (VFSS). It requires the patient to stay in a fixed position and consume barium-coated materials; X-ray videos are taken, usually on the sagittal plane. This modality, however, has a risk of excessive radiation exposure, resulting in low repeatability in clinical use [7,8]. The recent development of point-of-care ultrasound (POCUS) increases the possibility of using it to monitor swallowing function at the bedside [9]. Ultrasound imaging is a rising star as it is “simple and repeatable” and “gives real-time feedback” [10].

The hyoid bone excursion is a significant clinical indicator when using ultrasound for swallowing assessments. Hyoid has a uniform movement pattern in different individuals: on the sagittal plane, it first rises and then moves anteriorly to reach its maximum displacement, before returning to its original place [11,12,13]. This pattern can be classified as the elevation, anterior movement, and return phases. Gender [14] and age [11] might influence the duration that the hyoid spends in each phase and the moving distance. For example, age-related physiological changes can result in suprahyoid and infrahyoid muscle atrophy that reduces hyoid bone elevation [11]. The elderly also exhibit reduced maximum vertical and anterior hyoid bone movement [15], while the likelihood of having aspiration is 3.7 times greater for individuals who demonstrate reduced hyoid excursion [16]. Quantitatively, ultrasound can be used to make kinematical measurements of hyoid excursion. In previous literature that used manual annotation, hyoid movement analysis using ultrasound ranged from 1.3 cm to 1.7 cm in patients with different medical histories [17] and varied from 1.34 cm to 1.66 cm for different age groups [11]. Meanwhile, Hsiao et al. (2012) proved that hyoid bone displacements measured using submental ultrasonography and by VFSS have good correlations [17]. While mandible structures can sometimes block the structures of the hyoid bone in VFSS [18], it does not happen in ultrasound midsagittal images as the mandible is located away from the hyoid bone location at the midsagittal plane of the body.

Many studies used manual labeling [14,17,19,20] to identify the change in hyoid location, hyoid movement pattern, and maximum displacement during swallowing. Identifying the locations of the hyoid bone in ultrasound images is a laborious and time-consuming task. An automation process with the help of recent progress in computer vision and artificial intelligence would be preferred to reduce the time needed for reviewing the ultrasound video frame by frame. Lopes et al. (2019) used You Only Look Once version 3 (YOLOv3)to locate the hyoid bone in the ultrasound imaging [21], which gives some insights on automatically labeling the hyoid location in one single ultrasound image, yet they did not test tracking of hyoid bone locations in subsequent frames in ultrasound videos. Detection and segmentation-related deep learning methods have been applied to track the locations of the hyoid bone in VFSS [22,23]. However, tracking-related deep learning methods have not been applied to ultrasound videos. Therefore, we propose to use the state-of-the-art deep learning tracking algorithms (i.e., Fully-Convolutional Siamese Networks (SiamFC) [24], Accurate Tracking by Overlap Maximization (ATOM) [25], Discriminative Model Prediction (DiMP) [26]) and correlation filter tracking algorithms (i.e., Discriminative Correlation Filter Tracker with Channel and Spatial Reliability (CSRT) [27], Efficient Convolution Operators (ECO) [28,29]) to a new ultrasound swallowing video (USV) dataset. These methods could potentially reduce the requirement on manual assessment of 300–400 frames of ultrasound down to only one frame. After the hyoid bone’s location in the first frame of the ultrasound image series is annotated with a bounding box, tracking algorithms will then predict the bounding boxes in each subsequent frame to indicate the location of the hyoid bone.

## 2. Materials and Methods

### 2.1. Ultrasound Swallowing Videos (USV) Dataset

The USV were collected from 10 young and healthy adults (5 M + 5 F, aged 25.0 ± 2.6 years). They had no history of dysphagia, swallowing complaints, nor any craniofacial anomalies. Each subject performed five types of swallows in different volumes and consistencies: 5 mL and 10 mL of paste liquid, 5 mL and 10 mL of thin liquid, and dry swallow. Paste liquids were thickened to level 4 of the International Dysphagia Diet Standardisation Initiative (IDDSI) framework [30]. Ethical approval was obtained from the Human Subjects Ethics Sub-Committee (HSESC) of the Hong Kong Polytechnic University (HSESC Reference Number: HSEARS20191130001).

A convex ultrasound transducer (Aixplorer Multiwave Ultrasound System with an XC6-1 convex probe, Supersonics imagine, Aix-en-Provence, France), with a frequency bandwidth of 1–6 MHz, depth setting of 80 mm, in harmonic mode was placed in the midsagittal plane sub-mentally of the subject. The settings of the gain and time gain compensation (TGC) of the machine were set the same for all subjects. The subjects sat in an upright position. Two gel-pads (Acton^®^ BOL-I-X bolus with film, Action^®^, Hagerstown, MD, USA) of different dimensions 10 × 10 × 1 cm^3^ and 10 × 2 × 1 cm^3^ were placed at the submental area participants to ensure ultrasound coupling from the transducer to the subject as demonstrated in previous study [31]. The swallowing process was recorded with 32 frames per second (fps). Timestamps of the anatomical movement events were identified manually and recorded by trained speech therapists, such as the start of humming, the onset and offset of hyoid bone movement and the end of the swallow. Specifically, humming was used to check the alignment of the ultrasound probe and the anatomical center of the subject before the start of swallow. The hyoid onset and offset were the first frame when the hyoid bone moved forward and the frame when the hyoid bone started to return from its most anterior position. The end of a swallow is when the hyoid bone returned to its original position. The recordings were trimmed from humming to 50 frames after the end of a swallow, forming the final dataset with 50 videos and an average of 382 frames per video sequence.

Considering that the size of the hyoid bone is small, the location of the hyoid bone on every frame was annotated as a point by trained speech therapists to meet the clinical standard. In the ultrasound images, the hyoid was identified as “a high echoic area with a posterior acoustic shadow” [11]. Therefore, a point was placed at the intersection of the geniohyoid muscle and the superior border of the acoustic shadow, as shown in Figure 1. First, the annotation point was placed manually in every 5 or 10 frames with the help of interpolation mode of Computer Vision Annotation Tool (CVAT), then the points in each frame were revised and corrected to achieve frame by frame annotations. The size of each frame of USV is 720 × 540 pixels. Calibrated from the scale bar of the image, conversion between real distance to pixel is about 1 mm to 3.078 pixels, where the real anatomical size of 1 pixel was 0.325 mm. A bounding box of 30 × 30 pixels2 (~95 mm^2^), with a center at the ground truth annotation point, was given in the first frame to initialize tracking. The tracking algorithms would provide a bounding box in each subsequent frame (Figure 1), and the centers of those bounding boxes were considered to be the center locations of the hyoid bone. None of the ground truth point annotations in frames other than the first frame were used to generate bounding boxes; they were only intended to evaluate the performance of the tracking methods.

### 2.2. Algorithms for Hyoid Bone Tracking

Several state-of-the-art deep learning tracking algorithms and correlation filter tracking algorithms were applied to track the hyoid bone location in swallowing ultrasound. They were either known for superior performance in visual object tracking (VOT) challenges [32,33] or have reported a great performance gain. Most importantly, they are all real-time trackers, which would facilitate the clinical translation of evaluating ultrasound swallowing videos.

#### 2.2.1. Siamese Trackers

Siamese trackers use the same offline-trained backbones for the template branch and detection branch, in which the previous reference frame is served as a model template for the current frame. The pioneer, SiamFC (Figure 2), uses the fully convolutional network to extract features and computes the similarity between two image patches on a single dense grid, namely a score map, in one evaluation [24]. During tracking (inference), the exemplar patch and search patches of different scales are normalized to the exemplar image (127 × 127 × 3) and three search images (255 × 255 × 3) and fed to the same backbone. Exemplar and search feature maps are generated after passing the feature extraction backbone. Applying cross-correlation on the output exemplar feature map and three search feature maps would produce three score maps. Then up-sampling the three score maps by bicubic interpolation [34] could give three score maps with higher resolution. The peak response out of the three score maps was selected, and its relative distance away from the center represents the displacement of the hyoid from the previous frame to the current frame. During training, SiamFC uses weighted binary cross-entropy loss to optimize the results on the score map by minimizing the distance of the elements on the score map and the label matrix.

#### 2.2.2. Multi-Stage Trackers

Deep multi-stage trackers split the tracking task into coarse localization of the object, usually done by classification, and refined bounding box estimation, through methods like bounding box regression or Intersection over Union (IoU) prediction (Figure 3). ATOM [25] was the pioneer to break the tracking task into a classification branch and a target estimation branch. Online classification generates proposals close to the peak response in the score map by adding Gaussian or uniform random noises. Offline trained IoU Net [35] in the target refinement branch optimizes the coarse locations given by the proposals and produces a series of IoU scores for each initialized proposal bounding box. Averaging sizes and locations of the top three proposals, ranked in IoU scores, generates the predicted bounding box. During training, the IoU Net is optimized by gradient ascent with the help of precise region of interests (PrRoI) pooling layers. DiMP [26] improves the online classification to offline trained networks to extend the control over the tracking performance while not losing the discriminative power by integrating background appearance in the model prediction architecture. It uses hinge-like loss to distinguish the foreground from the background better and to ensure excellent classification performance.

#### 2.2.3. Correlation Filter Trackers

Discriminative correlation filter (DCF) trackers (i.e., CSRT [27] and ECO [28,29]) perform convolution between the target and the detection frame and train a filter online, at the same time as performing tracking in the Fourier domain to generate a response map (Figure 4) [36]. The filter localizes the target in the successive frame before being updated. The superior performances of correlation filter trackers can be attributed to the dense sampling achieved by circulantly shifting the target path samples, which profoundly augments the training data, as well as by using the element-wise product in the Fourier domain in place of the time domain convolution, to save tremendous computational power. CSRT (also named CSR-DCF) uses the channel reliability map to tune more adaptable spatial maps while training. It is implemented with an OpenCV Multitracker class in this study. ECO uses the VGG-M network (pre-trained on ImageNet) [37] to replace hand-crafted features and produce a multi-resolution (deep) feature map. It also adjusts C-COT’s iterative optimization strategy [38] to a sparser updating scheme to decrease the model complexity and save memory for remembering earlier frames. ECO is implemented with GitHub PyTracking repository in this study.

### 2.3. Implementation and Training Details of Deep Trackers

Out of the 50 sequences from five females and five males of five types of swallow each, 30 sequences from three females and three males were used to train the models; 10 sequences from one female and one male were used for validation; 10 sequences from one female and one male were used for the tests. All models were implemented in PyTorch, and they were trained on the Ubuntu 20.04 system with an Intel i5 CPU processor, 15G RAM, an NVIDIA RTX2060 GPU. During training, video sequences were uniformly selected; two images within 100 frames in one sequence, also called an image pair, were used to pass through the reference and test branch separately.

A few feature extraction backbones, including AlexNet [39], VGG 16 [40], ResNet 18 and 50 [41], and CIResNet 22 [42], were used in SiamFC, ATOM, and DiMP. Several trackers were selected for further evaluation, including SiamFC trained with AlexNet [43] and CIResNet22 from scratch, ATOM trained with ResNet 18 from scratch, ATOM finetuned from pre-trained ResNet 50, DiMP finetuned from pre-trained ResNet 18 and ResNet 50, since they have demonstrated a relative superior performance in the preliminary results of precision. The baselines used in SiamFC were slightly different from their original design due to the no padding specification in SiamFC. The details of how the baselines were revised can be found in [24] and [42]. Meanwhile, ATOM with ResNet 50 and DiMP with ResNet50 could be roughly compared with SiamFC with CIResNet 22, as ATOM and DiMP use ResNet block 1-3 as one of the feature extraction levels which is around the level of ResNet 22.

Data augmentation tricks, such as jittering on the center to crop images with offsets and stretching, were applied. All models had trained 50 epochs, and all checkpoints at 30 epochs were selected for evaluation according to the validation loss. Five thousand image pairs were used in one epoch to train SiamFC for 50 epochs with an SGD optimizer. The learning rate was annealed exponentially from 1 × 10^1^ to 1 × 10^5^ for SiamFC with AlexNet and from 1 × 10^2^ to 1 × 10^3^ for SiamFC with CIResNet 22. Eight thousand image pairs were used in one epoch to train ATOM. Adam [44] was used to optimize ATOM with ResNet 18 and ResNet 50 starting from learning rate 1e-1, with a gamma decay of 0.2 every 15 epoch. Six thousand image pairs were used in one epoch to train DiMP, with the settings listed in the PyTracking repo [26].

### 2.4. Evaluation

The evaluation of all trackers was performed on ten test video sequences, which were five types of swallowing clips collected from one female and one male. Given that (1) the center location of the hyoid bone in each frame was of more interest; and (2) the size of the prediction bounding box was less concerned; only evaluation metrics that compare the error between the predicted location (the center of the prediction bounding box) and ground truth annotation point were used to evaluate the data, including center error (1), precision (2), RMSE (3), and average error (4) [45,46,47,48].
(1)$$CE={\left\{\delta_{t}\right\}}_{t=1}^{N},\;\delta_{t}=\|x_{t}^{P}-x_{t}^{GT}\|=\sqrt{{\left(x_{t}^{P}-x_{t}^{GT}\right)}^{2}+{\left(y_{t}^{P}-y_{t}^{GT}\right)}^{2}}$$
(2)$$Precision=\frac{success}{total}=\frac{1}{N}\;\overset{N}{\underset{t=1}{\sum }}\delta_{t}\leq threshold$$
(3)$$RMSE=\sqrt{\frac{1}{N}\;\overset{N}{\underset{t=1}{\sum }}\delta_{t}^{2}}$$
(4)$$AE=\frac{1}{N}\;\overset{N}{\underset{t=1}{\sum }}\delta_{t}$$

In one sequence of *N* frames, we have a center error, which is the set of Euclidean distance between prediction and annotation at every frame; precision scores at different thresholds; a root-mean-square-error; and an average error.

The standard one-pass evaluation (OPE) was used for precision analysis, as zero reinitialization would most adequately simulate the case of the application reported in this study [46]. A frame would be considered as correctly tracked if its center error was within a distance threshold. The precision plot showed the percentage of correctly tracked frames for a range of distance thresholds. This curve was unambiguous and easy to interpret: A higher precision at low thresholds means the tracker is more accurate; high precision on a high threshold can indicate the robustness of the trackers. This is because robustness is defined as the tracker’s resilience to failures in challenging scenarios and its ability to recover, and a lost target would prevent the tracker from achieving perfect precision for a very high threshold range.

The Pearson correlation coefficient of the *x* and *y*-axis between ground truth and the inference of all frames was calculated to provide clinically relevant comparisons between the trackers. The range of motion (ROM) of the hyoid bone was calculated from hyoid onset to maximum displacement before hyoid offset, which represents the maximum elevation and anterior displacement of the hyoid during swallowing [49]. Furthermore, the relative error of ROM was calculated using Equation (5).
(5)$$\frac{\left|ROM\;of\;ground\;truth-ROM\;of\;inference\right|}{ROM\;ground\;truth}\times 100\%$$

## 3. Results

In the experiment, ten annotated test sequences were used for evaluating the tracker’s performance. The inferred hyoid locations were compared with the corresponding manual annotation in each video sequence. Appendix A showed an example case (female swallowing 10 mL of thin liquid), tracked with SiamFC (AlexNet). As visualized in the video, the traces of predicted hyoid and ground truth hyoid were moving similarly, and the locations of those two points were always staying close.

With the recorded timestamps of hyoid onset and offset, a comparison of the hyoid movement pattern between these two events of annotation and inference from an example case is shown in Figure 5. Ground truth hyoid onset locations are set as the origin of Figure 5a,c,d. In the 2D Cartesian plot Figure 5a, the two traces moved towards the left then vertically downwards, representing elevation and anterior movement of hyoid in anatomical displacement. Since the timestamp of hyoid onset did not happen at the first frame, the starting location of inference and ground truth were not the same in Figure 5a. For polar plot Figure 5b, both ground truth and inference locations of hyoid bone at hyoid onset were set at the origin of the plot for visualization of relative movement. The two traces moved in a similar angle and range of distance. The inference traces also stayed close to the ground truth traces in the *x*/*y*-axis coordinates Figure 5c,d. Thus, the prediction result, generated by SiamFC (AlexNet), had a comparable movement pattern and displacement to the ground truth one, i.e., manual annotation.

Analyzing the performance of the models quantitatively with a precision plot could reflect the accuracy and robustness of the model at different location error thresholds, as a way to verify whether the models could accurately extract the information of hyoid bone locations. It was considered that the model had comparable accountability to manual annotations if it had a high precision at an acceptable error threshold. The threshold was chosen at 10 pixels and 5 pixels, which are anatomical lengths of anatomical length 3.25 mm and 1.63 mm in the dataset, comparable to the measurement error reported using VFSS with human annotation of 2.62 to 2.89 mm [50]. As shown in Figure 6, SiamFC with AlexNet backbone achieved the highest mean precision of 98.9% at the threshold of 10 pixels and a mean precision of 80.5% at 5 pixels.

Quantitative analysis conducted over the 10 test sequences in full-length were shown in Table 1. The results of all trackers were real-time on an RTX 2060, though slower performance would be expected on an embedded system for portable ultrasound devices. Results from Table 1 suggested SiamFC had better performance in all analysis methods in terms of accuracy and speed (175 fps). SiamFC trained with USV gave an RMSE of 3.85 pixels ± 1.06 pixels (1.25 mm ± 0.34 mm) and an AE of 3.28 pixels ± 1.10 pixels (1.07 mm ± 0.36 mm). This result appears to outperform a reported RMSE of 3.2 mm ± 0.4 mm in a previous study using deep learning trackers on VFSS [23].

Good correlations were shown between ground truth and inference location with a Pearson’s correlation coefficient of 0.985 ± 0.013 and 0.919 ± 0.034 on the *x* and *y*-axis, respectively. The relative error of ROM was 9.5% ± 6.1%, compared to the relative error of 3.3% to 9.2% reported in the previous study [49].

The precisions at both thresholds of 10 pixels and 5 pixels were tested to explore the case of a possible stricter system. In the precision plot, the results at 10 pixels were quite convincing. However, the standard deviation was dramatically increased at the threshold of 5 pixels. The high standard deviation indicated that the performances for different frames vary, and there could be outliers existing in some frames.

## 4. Discussion

In this study, we proposed to use deep learning tracking algorithms and correlation filter tracking algorithms to automatically track the locations of the hyoid bone in swallowing clips collected using ultrasound imaging. Generally, SiamFC trackers outperform ATOM, DIMP, CSRT, and ECO. This could be attributed to the fact that hyoid bone tracking in ultrasound images has a relatively simple background and contains no distractor with similar features; it can also be attributed to the reason that only the center location of the tracking box is concerned and tested. This minimum requirement in such a task could make the proposal refinement step in ATOM and DiMP overcomplex for this task. Meanwhile, Siamese trackers contain no online learning parts, which ensures their speed performance. For such a tracking task, deeper feature extraction backbones did not have significant performance gain but instead slowed the tracking process.

Overall, SiamFC has a superior tracking performance. This method could facilitate speech therapists to perform routine evaluations on patients’ swallowing conditions using ultrasound imaging by replacing manual annotations frame by frame with automatic tracking. The method was proven to have reliable performance qualitatively with visualized traces and quantitatively with precision, RMSE, AE, Pearson correlation, and relative error of ROM. Although the subject group in this study might be different from other studies, the center error seems to be comparable with the manual measurement error of VFSS and could be smaller than other automatic hyoid tracking methods on VFSS.

While the relative error of ROM suggests good agreement between inference and ground truth, the *y*-axis ROM showed a more significant relative error in all tracking methods. Pearson correlation of center location was also lower in the *y*-axis. This could be due to: (1) acoustic shadow in the ultrasound image vertically above the hyoid bone and (2) reduced range of motion in the *y*-axis due to ultrasound probe compression on the gel pads and the tissue.

As mentioned previously, the model had a high standard deviation at the threshold of 5 pixels (1.63 mm), so we assessed possible outliers using the test sequence of the largest RMSE (5.95 pixels) and AE (5.70 pixels). As shown in Figure 7, we concluded that the sequence had a higher error than other sequences because the prediction location was always on the left of ground truth, which generated a higher systematic error. Meanwhile, the figure demonstrated outliers around frame 542 for a short range of time. While the tracker chose the location left to acoustic shadow, ground truth was at the middle acoustic shadow.

To analyze the possible cases where the model failed expectation, the frames in the test data where center error exceeded 10 pixels were also examined. It was found that large center errors existed when hyoid movement speed was higher. As ultrasound imaging requires the line-by-line acquisition of reflected sound waves to form an image, objects could repeatedly exist in different locations in one frame when objects move faster than image acquisition lines. Figure 8 shows an ultrasound frame where two acoustic shadows can be observed as the hyoid bone exists in both locations during fast movement. This could be solved by using ultrasound imaging with higher frame rates, such as ultrafast ultrasound, or by introducing a velocity smoothing function to find the highest possible location of hyoid bone during the fast-moving frames. This might also be addressed with model training approaches, such as applying a higher weight in loss function if one image in the pair is close to hyoid movement events.

This study has several potential future directions. Training of detection algorithms, such as YOLO [51], Faster-RCNN [52], and SSD [53] can be added in the first frame to complete the entire automation process. Algorithms could also be developed to distinguish the hyoid movement events such as hyoid onset and hyoid offset. Multiple object tracking can be used to detect the absolute or relative distance between the hyoid bone and the laryngeal cartilage. Besides, segmentation algorithms may be used to discern other anatomical landmarks like geniohyoid muscles and tongue.

A limitation of the study is that it has only included young and healthy adults, considering elderly and dysphagia subjects might have different movement patterns. Another limitation is that the number of trials was also not enough for comparisons of performance between swallowing of different bolus types. Since only ten videos from one male and one female, five bolus types each, were used for testing, the data size was not enough for statistical analysis of tracker’s performance on types of swallow. Future direction will need to include a larger dataset, i.e., older adults or dysphagia patients, or more videos per type of swallows, to enhance the tracking algorithm’s applicability to broader population groups.

## 5. Conclusions

In this work, we tested the performance of state-of-art deep learning algorithms and correlation filters on tracking the hyoid bone location in ultrasound swallowing videos. The performance of SiamFC in quantitative analysis methods was superior to other methods tested in terms of speed and accuracy. It had comparable performance with the manual annotation and could serve as a powerful tool to relieve the clinical practitioners from reviewing hyoid locations frame by frame tediously in ultrasound images.

The precision of this method is 98.9%. RMSE and AE, suggesting the error of the tracking method is around 1.07 mm to 1.25 mm. The tracker has also demonstrated accurate results in ROM with a relative error of 9.5% ± 6.1%. The results have shown that the tracker has comparable performance with human annotation in our USV dataset and comparable measurement error of hyoid bone on VFSS. This approach could also possibly outperform other hyoid tracking methods on VFSS with lower RMSE.

## Figures and Tables

**Figure 1 sensors-21-03712-f001:**
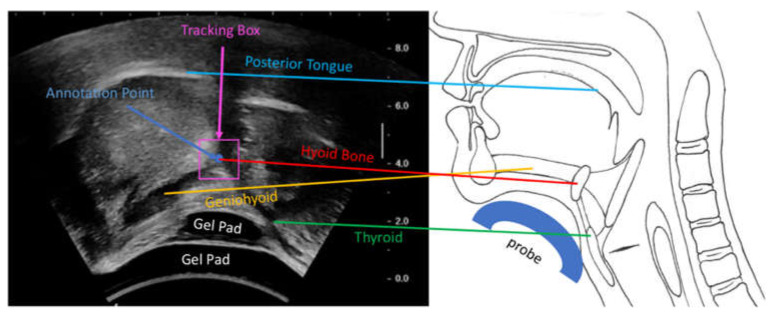
The left side of the figure shows an example ultrasound image with labeled anatomical structures, as illustrated on the right side. The hyoid bone annotation point was placed at the intersection of the geniohyoid muscle (left) and the acoustic shadow (above). During inference, a bounding box tracked the hyoid bone location.

**Figure 2 sensors-21-03712-f002:**
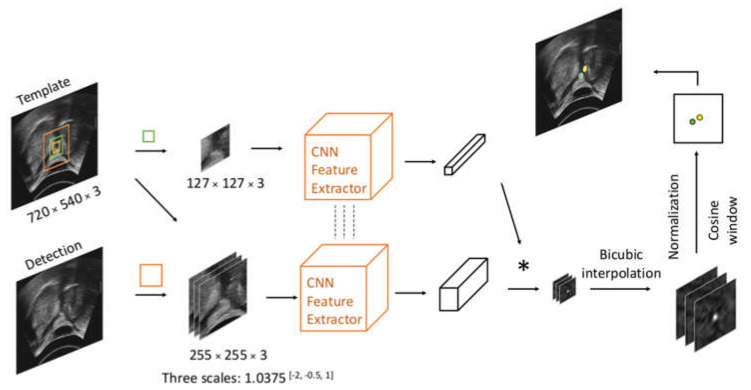
Workflow of Siamese trackers. The center of the yellow bounding box indicates the hyoid location in the last frame; the green box and orange box centered on the last frame hyoid location will crop exemplar patch and search patch on the previous frame and current frame, respectively. The green point indicates the peak response on the score map, while the yellow one denotes the center location of the current frame.

**Figure 3 sensors-21-03712-f003:**
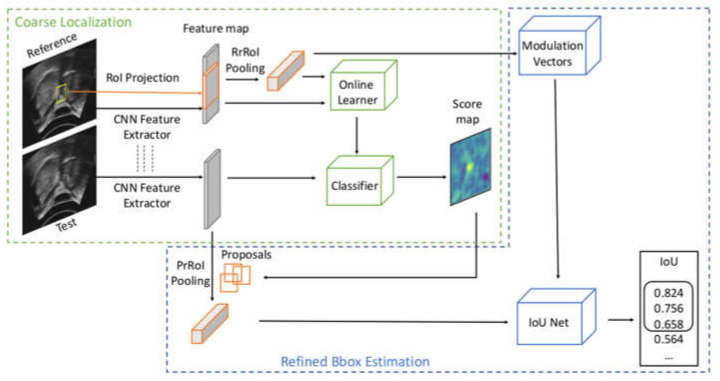
Workflow of multi-stage trackers. Precise region of interests (PrRoI) pooling layers [35] can convert features of different sizes into the same size while enabling the computation of the gradient of Intersection over Union (IoU) with respect to the bounding box coordinates. IoU Net outputs IoU scores for each proposal, and the top three ranked proposals are averaged to produce a robust prediction bounding box location.

**Figure 4 sensors-21-03712-f004:**
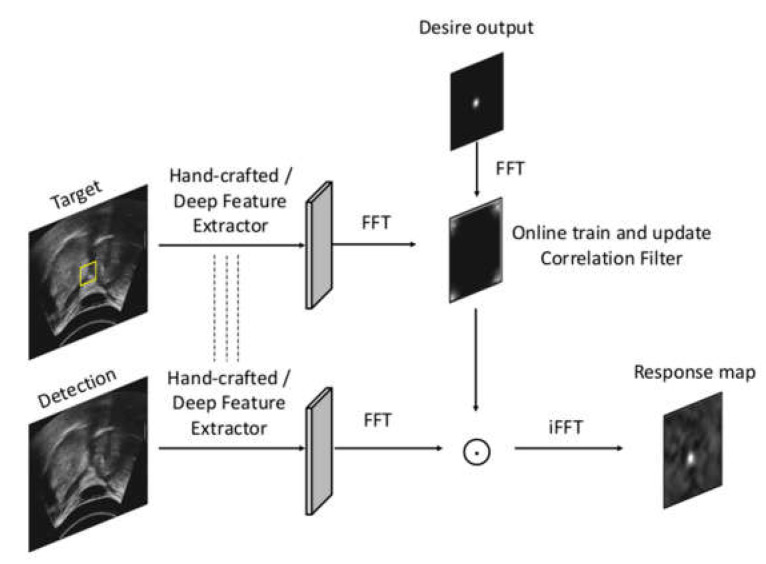
Workflow of correlation filter trackers.

**Figure 5 sensors-21-03712-f005:**
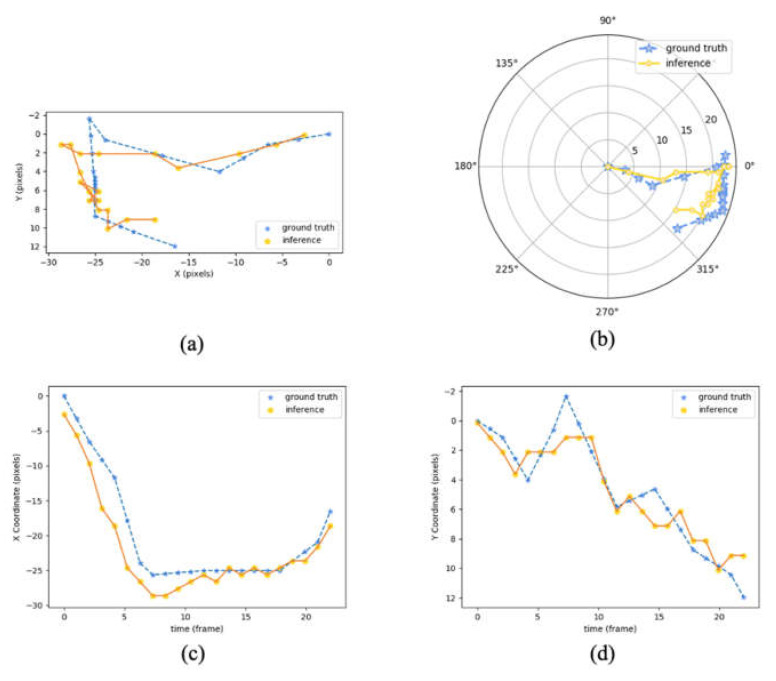
Hyoid center trace plots between the timestamps of hyoid onset (the frame when the hyoid starts to move) to offset (at the moment when the hyoid starts to move away from its maximum position of superior-anterior movement) in 2D Cartesian axis (**a**), polar axis (**b**), *x*-axis (**c**), and *y*-axis (**d**). From an example test sequence of 10 mL thin liquid swallow, female subject, from hyoid onset to offset. The blue line represents the ground truth of the hyoid path, and the yellow one represents the inference path. Length unit at the polar axis is in pixels.

**Figure 6 sensors-21-03712-f006:**
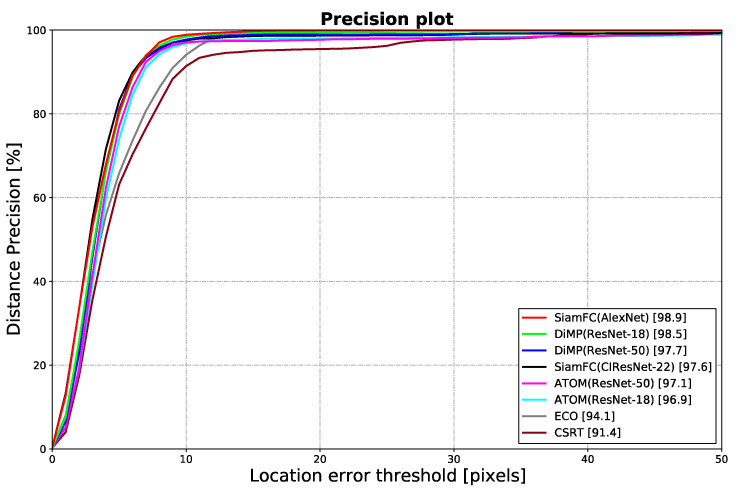
Precision plot shows the mean distance precision of 10 test sequences in full-length at different location error thresholds. The legend shows the precisions of different trackers at the threshold of 10 pixels.

**Figure 7 sensors-21-03712-f007:**
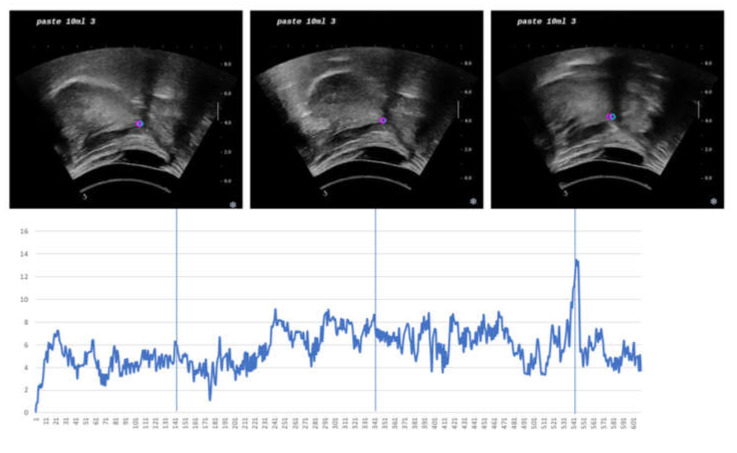
Performance plot. Center error of all frames from a test sequence in full-length where the female swallows 10 mL of paste liquid. The *y*-axis is a center error in pixel and the *x*-axis is frame number. Three example images from every 200 frames were chosen and displayed above the plot. The pink dot represents the center of inference, and the blue dot represents the center of ground truth (annotation).

**Figure 8 sensors-21-03712-f008:**
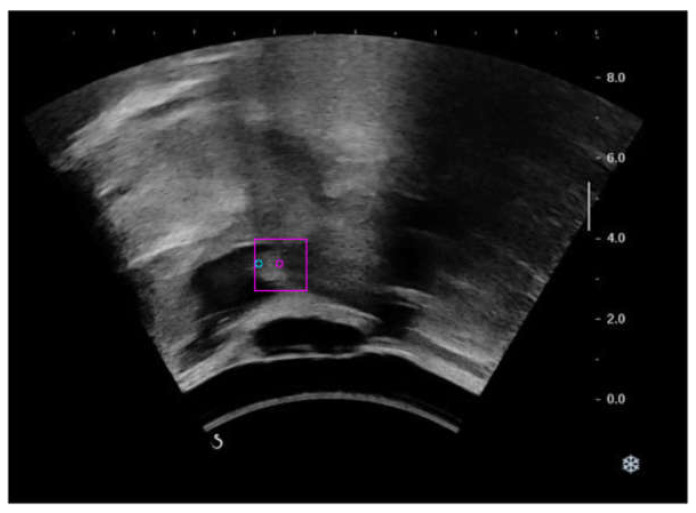
A frame in which two acoustic shadows of hyoid bone were seen due to fast hyoid movement speed. Ground truth location (blue dot) and prediction (pink dot with pink bounding box).

**Table 1 sensors-21-03712-t001:** Results of quantitative evaluation between ground truth hyoid locations and hyoid locations of 10 test sequences in full-length: Precision at 5 and 10 pixels, RMSE, AE, Pearson correlation on *x*-axis and *y*-axis, relative error of ROM in the *x*-axis, *y*-axis and straight-line distance, and tracker frame rate.

Tracking Methods	Precision at 10 Pixels ± SD (%) ↑	Precision at 5 Pixels ± SD (%) ↑	RMSE ± SD (Pixel) ↓	AE ± SD (Pixel) ↓	Pearson Correlation *x*-Axis ↑	Pearson Correlation *y*-Axis ↑	Relative Error of ROM in *x*-Axis (%) ↓	Relative Error of ROM in *y*-Axis (%) ↓	Relative Error of ROM (%) ↓	Tracker Frame Rate (FPS) ↑
SiamFC (AlexNet)	98.9 ± 1.7	80.5 ± 18.9	3.85 ± 1.06	3.28 ± 1.10	0.985 ± 0.013	0.919 ± 0.034	13.3 ± 9.6	67.4 ± 70.1	9.5 ± 6.1	175 ± 2
DiMP (ResNet-18)	98.5 ± 3.3	79.9 ± 18.20	4.66 ± 2.24	3.65 ± 1.29	0.980 ± 0.013	0.883 ± 0.102	12.8 ± 8.2	69.8 ± 34.1	11.2 ± 7.7	63 ± 2
DiMP (ResNet-50)	97.7 ± 5.5	81.1 ± 15.6	4.95 ± 3.13	3.87 ± 1.61	0.979 ± 0.016	0.890 ± 0.123	14.4 ± 12.9	81.5 ± 85.4	14.4 ± 10.2	48 ± 1
SiamFC (CIResNet-22)	97.6 ± 3.2	83.2 ± 17.0	5.21 ± 3.59	3.64 ± 1.54	0.951 ± 0.109	0.735 ± 0.424	34.1 ± 83.1	157.5 ± 228.2	35.5 ± 90.8	116 ± 7
ATOM (ResNet-50)	97.1 ± 3.6	77.0 ± 18.8	7.93 ± 5.95	4.78 ± 1.81	0.910 ± 0.145	0.751 ± 0.243	28.8 ± 32.2	227.2 ± 391.3	21.2 ± 29.6	32 ± 2
ATOM (ResNet-18)	96.9 ± 3.4	74.0 ± 19.7	7.36 ± 4.35	4.71 ± 2.05	0.956 ± 0.061	0.734 ± 0.212	56.3 ± 84.7	229.6 ± 361.3	52.0 ± 89.8	43 ± 1
ECO	94.1 ± 12.8	65.8 ± 30.6	5.16 ± 2.16	4.43 ± 2.10	0.978 ± 0.021	0.890 ± 0.083	191 ± 18.7	150.4 ± 238.5	17.4 ± 13.6	24 ± 3
CSRT	91.4 ± 9.4	63.1 ± 25.3	8.23 ± 5.19	5.90 ± 2.75	0.922 ± 0.116	0.710 ± 0.263	27.6 ± 26.4	93.0 ± 100.9	26.9 ± 25.3	61 ± 3

↑ ↓: Arrow pointing up indicates larger value is preferred and arrow pointing down indicates smaller value is preferred.

## Data Availability

The raw ultrasound videos, annotations, trained networks and tracking results presented in this study are available on request from suri.d.feng@connect.polyu.hk.

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
