# Peer review of "Automatic Hyoid Bone Tracking in Real-Time Ultrasound Swallowing Videos Using Deep Learning Based and Correlation Filter Based Trackers"

_sensors, 2021, doi:10.3390/s21113712_

Round 1

Reviewer 1 Report

The paper present the implementation and evaluation of tracking algorithms based on deep learning and correlation filters for tracking the movement of the hyoid bone on submental ultrasound videos. This work is an original exploration and represents a very interesting approach for assessing swallowing disorders. The manuscript is well written and results are clear. I only have some minor observations and suggestions:

  • Although the title, abstract, and introduction suggest an unbiased evaluation of the mentioned tracking algorithms, methods and conclusions are biased towards SiamFC as if the goal of the work were to propose it (like a new method), and merely use the other methods as benchmark. If substantial functional modifications to the SiamFC baseline architecture were made, then it would be important to highlight them and add experiments that demonstrate how those modifications influence the tracking performance. For what I read, it seems that modification were mainly for adapting to the application and specifics of the dataset, with many implementation details in common with all other algorithms. Otherwise, it's better to present each algorithm in the same way (e.g. given a subsection to each), focusing on the details required for the specific implementation given the application and dataset, and then presenting common implementation and training details (for deep learning methods). If following the last, conclusions should be revised for not presenting SiamFC as the "proposed algorithm" but as that which shows the best results.
  • In line 206 is stated that "ATOM and DiMP, were used to compare with SiamFC as they are openly available" as if the implementation of SiamFC were not also open available. It may say "they are also openly available". 
  • Regarding SiamFC implementation, if its open source (GitHub) implementation on Tensorflow was used it should be specified. In any case, it's better to specify the (Python?) frameworks used in the implementation. 
  • There are no implementation details for the correlation filter trackers (source?, libraries?) 
  • There is an obvious (to me) future direction of the research that is not addressed in the discussion: include old and/or non-healthy subjects in the study. In fact, the statement in line 304 that affirm that the explored methods "can have matching performance to manual annotation in the gold standard VFSS" may not be always true since the main target situation has not been tested (actually this affirmation should be in the discussion section).
  • The insertion of Figure 2 with the details of AlexNet is difficult to read. Consider put it in another figure.
  • In Figure 3 it's not clear if trace plots starts at frame 0 (initial frame). If so, why the initial X coordinates of inference and ground truth are not the same? 
  • Some language suggestions: in line 31 "has increased to" may be replaced with "reaching"; in line 51-52 "Elderly person exhibited" may be replaced with "Elderly person exhibit"; In line 66 "yet" should be omitted; in line 68 consider a better word for "human power"; in line 75 "proposed" should be replaced with "propose"; in line 77 "our ... dataset"  may be replaced with "a new ... dataset"; in line 80 "was annotated using bounding box" may be replaced with "is annotated with a bounding box".

Author Response

Point 1: Although the title, abstract, and introduction suggest an unbiased evaluation of the mentioned tracking algorithms, methods and conclusions are biased towards SiamFC as if the goal of the work were to propose it (like a new method), and merely use the other methods as benchmark. If substantial functional modifications to the SiamFC baseline architecture were made, then it would be important to highlight them and add experiments that demonstrate how those modifications influence the tracking performance. For what I read, it seems that modification were mainly for adapting to the application and specifics of the dataset, with many implementation details in common with all other algorithms. Otherwise, it's better to present each algorithm in the same way (e.g. given a subsection to each), focusing on the details required for the specific implementation given the application and dataset, and then presenting common implementation and training details (for deep learning methods). If following the last, conclusions should be revised for not presenting SiamFC as the "proposed algorithm" but as that which shows the best results.

Response 1: Thank you very much for the guidance on paper structure. It is really helpful for us in learning manuscript writing. The method part and result part have been revised accordingly. We have allocated subsections to all each type of tracking algorithms: instead of focusing on SiamFC, all other types of tracking algorithms are elaborated with roughly the same weight. Meanwhile, based on your suggestions, experiments on modifying baselines have been added. It has been also presented in a clearer way in method, result, and conclusion.

Point 2: In line 206 is stated that "ATOM and DiMP, were used to compare with SiamFC as they are openly available" as if the implementation of SiamFC were not also open available. It may say "they are also openly available". 

Response 2: Thank you very much for you comment. We agree this statement was very misleading. We have deleted this sentence. We have rearranged the order of presenting the three deep learning methods. Reasons of choosing SiamFC, ATOM, and DiMP are their real-time tracking performance and availability. This has been added in p.4 line 143

Point 3: Regarding SiamFC implementation, if its open source (GitHub) implementation on Tensorflow was used it should be specified. In any case, it's better to specify the (Python?) frameworks used in the implementation. 

Response 3: Thank you for your suggestion. The implementation of SiamFC in PyTorch has been added (p.6 line 227), the architecture is following the logic of: citation [42] huanglianghua, siamfc-pytorch, (2020), GitHub repository, https://github.com/huanglianghua/siamfc-pytorch

Point 4: There are no implementation details for the correlation filter trackers (source?, libraries?) 

Reponse 4: Thanks for your suggestion, the source and libraries have been added.

CSRT: opencv multitracker class p.5 line 206

ECO: GitHub pytracking framework p.5 line 111, citations [28] [29]

Point 5: There is an obvious (to me) future direction of the research that is not addressed in the discussion: include old and/or non-healthy subjects in the study. In fact, the statement in line 304 that affirm that the explored methods "can have matching performance to manual annotation in the gold standard VFSS" may not be always true since the main target situation has not been tested (actually this affirmation should be in the discussion section).

Response 5: Thank you very much for your suggestions. We have included your suggestion about limitation and future direction of this study in the discussion section (p.12 line 426). And yes, the affirming statement should not be in the results section, we have removed this statement from the results section (p.8 line 323).

Point 6: The insertion of Figure 2 with the details of AlexNet is difficult to read. Consider put it in another figure.

Response 6: Thank you for pointing out the figure drawing issue. AlexNet scheme has been deleted and Figure 2 has been modified.

Point 7: In Figure 3 it's not clear if trace plots starts at frame 0 (initial frame). If so, why the initial X coordinates of inference and ground truth are not the same? 

Response 7: Thank you very much for your comment. The trace plot in Figure 3 (Figure 5 in the revised manuscript) started from onset of hyoid movement, which was different from the starting frame (start of humming). Hence the frame 0 coordinates of annotation and tracking in the trace plots were different. However, we have initiated the starting coordinates in the polar axis plot only as a relative motion plot. Where (a), (c), (d) gives the actual coordinates in the frame, to illustrate the center error of the tracker, (b) gives a relative motion in polar axis for visualization of numerical parameters such as the range of motion. We have modified the caption of figure by adding ‘hyoid onset (the frame when the hyoid starts to move)’. The definition of timestamps such as hyoid onset is at p.3 line 106.

Point 8: Some language suggestions: in line 31 "has increased to" may be replaced with "reaching"; in line 51-52 "Elderly person exhibited" may be replaced with "Elderly person exhibit"; In line 66 "yet" should be omitted; in line 68 consider a better word for "human power"; in line 75 "proposed" should be replaced with "propose"; in line 77 "our ... dataset"  may be replaced with "a new ... dataset"; in line 80 "was annotated using bounding box" may be replaced with "is annotated with a bounding box".

Response 8: Thank you very much for pointing out our language mistakes. We have corrected according to all your suggestions, and some other grammatical issues.

Reviewer 2 Report

Major:

  • (lines 107 -108) The authors said that the end of swallowed was defined by the hyoid bone returned to its original position. However, in Figure 3 (c) and (d), at the end of the traces (i.e., frame #22), the traces of x and y axes did not look like back to its original position (i.e., x≠0; y≠0 at frame #22). Is this figure meant to show the traces of hyoid bone movement from onset to offset? Please explain. 
  • (lines 88) The authors examined the US swallowing videos with different types: paste liquid, thin liquid, and dry swallow. In the Results section, the tracker’s performances were evaluated without separating different swallowing types. Figure 5 and the paragraph (lines 352-362) suggest that thick liquid may induce different effects (i.e., errors and outliers). Please justify the evaluations were not performed by different swallowing types. If possible, please do the evaluations separately and discuss accordingly.

Minor:

  • (line 138) “from the previous from” —> “from the previous frame”

Author Response

Point 1: (lines 107 -108) The authors said that the end of swallowed was defined by the hyoid bone returned to its original position. However, in Figure 3 (c) and (d), at the end of the traces (i.e., frame #22), the traces of x and y axes did not look like back to its original position (i.e., x≠0; y≠0 at frame #22). Is this figure meant to show the traces of hyoid bone movement from onset to offset? Please explain. 

Response 1: Thank you very much for your comment. Figure 3 (Figure 5 in the revised manuscript) shows the hyoid movement pattern from the hyoid onset (the frame when the hyoid starts to move) and the hyoid offset (the frame when the hyoid starts to move away from its maximum position of superior-anterior movement). To clarify this point, caption of the figure has revised by adding “offset (at the moment when the hyoid starts to move away from its maximum position of superior-anterior movement)”. The definition of timestamps such as hyoid offset is at p.3 line 106.

Point 2: (lines 88) The authors examined the US swallowing videos with different types: paste liquid, thin liquid, and dry swallow. In the Results section, the tracker’s performances were evaluated without separating different swallowing types. Figure 5 and the paragraph (lines 352-362) suggest that thick liquid may induce different effects (i.e., errors and outliers). Please justify the evaluations were not performed by different swallowing types. If possible, please do the evaluations separately and discuss accordingly.

Response 2: Thank you very much for your comment. We have investigated on the results again, and higher inaccuracy did not happen in all cases when swallowing thick paste and larger volume (10ml), the example case chosen could be only due to a systematic error, where the inference bounding box is always on the left to the ground truth. We think we have made a mistake in suggesting the inaccuracy could be attributed to thick liquid. The sentence “This inaccuracy could be attributed to the swallowing of thick liquid consistency” has been deleted in discussion section (p.11 line 385). However, since only 10 videos (2 subjects of 5 swallowing types) were used in testing, the number of videos were not enough to perform meaningful comparisons on the tracker’s performance when swallowing different bolus types. Such comparison is recommended in the section “future direction and limitation of this study” (p.12 line 430).

Point 3: (line 138) “from the previous from” —> “from the previous frame”

Response 3: Thank you for pointing this out and requiring us to perform extensive English revisions. As suggested, the typing mistake has been corrected. A number of other typos and grammatical errors are also corrected in the revised manuscript. Meanwhile, we have rearranged the structure of the paper in a clearer logic especially in the method and result parts. We hope this would be easier for you to read.

Round 2

Reviewer 1 Report

I congratulate authors for the work done. Modifications improved the readability of the manuscript and usefulness for readers. 

Reviewer 2 Report

The authors have responded all my comments and revised the manuscript accordingly.